# Sociocultural and patient-health care professional related factors influencing self-management of multiethnic patients with multimorbidities: A thematic synthesis

Ahtisham Younas[1]*, Sharoon Shahzad[2], Clara Isabel Tejada-Garrido[3], Esther Nyangate Monari[1], Angela Durante[4]

1 Memorial University of Newfoundland, Saint John's, Canada, 2 Islamabad Nursing College, Islamabad, Pakistan, 3 Research Group in Care GRUPAC, University of La Rioja, Logroño, Spain, 4 Università del Piemonte Orientale, Vercelli, Italy

* ay6133@mun.ca

**Data Availability Statement:** No Original data were used during the preparation of this manuscript.

## Abstract

Self-management is critical for the general well-being and disease management of individuals with multimorbidities. A better understanding of sociocultural and patient-professional level factors affecting self-management can be valuable for designing individual and community-based strategies to promote optimal self-management. The purpose of this review was to explore sociocultural and patient-health care professional related factors affecting self-management among patients with multimorbidities. A metasynthesis was conducted. Literature was searched in PubMed, CINAHL, Scopus, Web of Science, and OVID databases. In total, 21 qualitative studies published from January 2010-March 2023 were critically appraised and reviewed. Thematic synthesis was used for analysis and eight descriptive and three analytical themes were generated. The analytical themes illustrated that personal and structural vulnerabilities, social and family struggles, and fragmented interpersonal relationships with health care professionals affect health care access, navigation, and self-management of individuals with multimorbidities. Engagement in self-management for individuals with multiple chronic conditions is hampered by under-resourced community and health care environments, structural vulnerabilities, familial and interpersonal conflicts, and disjointed relationships. There is a dire need to ensure optimal community resources to support individuals to address and navigate complexities associated with accessing care and effectively managing their illnesses.

## Introduction

Multimorbidity, defined as two or more chronic conditions, is an emerging issue for individuals and health care systems globally [1]. Individuals with multimorbidities require continuous self-management of their diseases to avoid worsening their general well-being [2, 3]. Self-management in chronic illnesses refers to "the intrinsically controlled ability of an active, responsible, informed and autonomous individual to live with the medical, role and emotional

This is a metasynthesis and the studies included in the review are already presented in the literature summary tables.

**Funding:** The authors received no specific funding for this work.

**Competing interests:** The authors have declared that no competing interests exist.

consequences of his chronic condition(s) in partnership with his social network and the healthcare provider" [4, p.10]. Self-management is critical for individuals with multimorbidities. These individuals need to frequently visit general health settings and specialized care units to screen for complications, assess ongoing and emerging needs, and seek guidance on their self-management plans [4, 5]. However, the ability of these individuals to seek health care guidance and engage in optimal self-management is affected by personal, interpersonal, socio-cultural, and health system-related factors [3]. Reviews synthesized outlined experiences of individuals with multimorbidities, nature of self-management, priorities of self-management, and individuals' experiences with health care processes. Two reviews [3, 6] highlighted some personal, interpersonal, and system-level issues encountered by these patients when striving for self-management. However, there is a gap in knowledge about critical sociocultural and patient-health care professional related factors affecting self-management of patients with multimorbidities. Gaining this understanding is essential to tackle contextual social and cultural factors impacting self-management and develop strategies to assist these individuals in performing optimal self-management.

## Background

### Burden of multimorbidity

The increasing incidence of multimorbidity leads to a global burden [7]. An estimated relevance of multimorbidity in high and middle-income countries, based on a review of 76 studies, is 33.1% with a pooled prevalence of 37.9% in high-income countries and 29.7% in low and middle-income countries [8]. Recently, Abebe et al. [9] noted an increased prevalence of multimorbidity in low and middle-income ranging from 3.2% to 90.5%. Multimorbidity increases with age and the number of chronic conditions [9, 10]. Ofori-Asenso et al. [10] estimated an overall prevalence of 66.1% in older adults. They also noted that, when a definition of three or more chronic conditions was used, 56.5% of individuals were considered to have multimorbidity. The increased prevalence of multimorbidity is associated with worst health outcomes, increased use of health care resources, and reduced functional and system capacity [11].

### Importance of self-management in multimorbidity

Self-management can play an instrumental role in enhancing the physical and psychological well-being and functioning of individuals with multimorbidities, thereby reducing deteriorating health outcomes and improving functional and system-level capacity. Fortin et al. [12] and Contant et al. [13] demonstrated that a multifaceted self-management intervention (i.e., education about nutrition, lifestyle, motivation interviewing, and counseling) among individuals with multimorbidities improved their health-directed behaviors, emotional well-being, self-monitoring, constructive attitudes, self-management skill and technique acquisition, and health services navigation. Dineen-Griffin [14], after a systematic review of 58 studies, noted that self-management has the potential to improve clinical outcomes, quality of life, self-efficacy, confidence, and disease control for individuals. The self-management interventions focused on providing knowledge about the chronic conditions, improving patient role in lifestyle changes, psychological coping, problem-solving and/or decision-making skills, medication adherence, developing self-management plans, and keeping self-monitoring logs. Of 58 studies, nine studies found a positive impact of self-management interventions on disease-specific outcomes, four studies noted improved self-efficacy, eight reported improved health-related quality of life, three studies demonstrated improved physical and social functioning, and 11 studies found a positive effect on psychological functioning.

### Reviews of literature on self-management and multimorbidity and gaps

Previous reviews focused on self-management in older people living with cancer and multimorbidities [15], characteristics of self-management in individuals with multimorbidities [3], priorities of individuals with multimorbidities [16], lived experiences of multimorbidity and managing physical and mental health issues [17], and experiences with multimorbidities and the health care processes [6]. Two of these reviews outlined some personal, interpersonal, and health care system-related issues affecting the self-management of individuals with multimorbidities. For example, van der Aa et al. [6] discussed that these individuals fail to receive holistic care due to the issues surrounding communication and relationship with health care professionals, lack of advocacy, health care access, and resources (interpersonal and system-level issues). Gobeil-Lavoie et al. [3] elaborated that individuals with multimorbidities focus on managing one condition at a time and do not receive adequate information from health care professionals on disease management (interpersonal issues). This, in turn, affects their emotional and physical health and makes them susceptible to complications. Self-management can have many physical, psychological, and disease-related outcomes in patients with multimorbidities [14]. While existing reviews offer some insights about factors affecting the self-management of individuals with multimorbidities, no reviews have synthesized the sociocultural and patient-health care professional related factors affecting the self-management of multiple chronic conditions.

## Purpose

The purpose of this review is to explore sociocultural (e.g., ethnicity, gender, cultural orientation) and patient-health care professional related factors (e.g., interpersonal relationship and communication) affecting self-management among individuals with multimorbidities.

## Methods

We used metasynthesis because it is a valuable approach for understanding experiences, perspectives of individuals about complex phenomena [18] through summating and, or aggregating findings from qualitative studies [19] Metasynthesis allows for generating a deep and broad understanding of qualitative literature on a particular topic by creating 'integrations that are more than the sum of parts, in that they offer novel interpretations of findings [20, p. 1358] We followed the PRISMA guidelines for reporting the metasynthesis [21] (S1 Checklist).

### Literature search

An exhaustive literature search [19] was performed in five databases to identify studies published from January 2010 until March 2023. The literature search was limited to these years to capture more contemporary literature on the topic. The databases included: Cumulative Index to Nursing and Allied Health Literature (CINAHL) (n = 335), PubMed (n = 580), Scopus (n = 501), Web of Science (n = 207), and Ovid (Medline) (n = 403) using the indexed terms, keywords, subject headings, and MESH terms. These search terms and MESH headings were first identified from PubMed and compared with the search terms from other databases to choose the most pertinent terms and subject headings. The terms included: "sociological factors,", "sociocultural determinants", "patient perspectives", "patient experiences", "multimorbidity*", "comorbidity", "multiple chronic conditions", "health care access", "health care services", "health care accessibility", "self-management*", "self-care", "qualitative", "lived experiences", "perspectives", and "professional-patient relations*". Additionally, terms and

keywords related to qualitative research methodologies such as "phenomenology", "narratives", "case study", "grounded theory", "exploratory qualitative", and "descriptive qualitative" were also used. The tiab-terms and truncation strategies were combined with names of qualitative study designs and Boolean variables "OR" and "AND" for advanced search. Search strategy for PubMed is presented in S1 Table.

## Inclusion and exclusion criteria

The inclusion criteria were: a) empirical research published from January 2010 to March 2023 in the English language in peer-reviewed journals, b) original qualitative studies including case studies, narratives, phenomenology, grounded theory, ethnography, descriptive qualitative, exploratory qualitative, and interpretive descriptive designs, c) qualitative studies including individuals with at least two or more chronic conditions as the target population and sample, d) studies that explored the lived experiences of self-management or self-care, and e) studies about individuals experiences and perspectives about social and cultural factors affecting their self-management. The exclusion criteria included: a) qualitative, quantitative, and mixed methods studies focused on sociocultural determinants and factors and their influence on health care access for patients with chronic diseases, but without multimorbidities, b) quantitative and mixed methods studies about sociocultural determinants affecting self-management in individuals with one or more chronic conditions, and c) literature reviews, discussion papers, dissertations, commentaries, editorials, and opinion pieces about the very topic.

## Search outcomes

In total, 2032 articles were identified from all the databases. After manually removing duplicates (n = 971), 1061 articles were screened by reading the abstracts and titles. Two independent reviewers (AY & SS) screened the titles and abstracts. Of these articles, only 31 articles were relevant for full-text screening. These records were sought for retrieval, and all of these records were successfully retrieved. These 31 articles were subjected to full screening, but only 21 met the inclusion and exclusion criteria. Fig 1 Literature Search provides the detailed search outcomes using the PRISMA diagram.

## Data extraction

Data were extracted using the summary tables in Excel. The tables included information about authors, country, purpose, study design, setting, sample characteristics, sampling technique, method of data collection, methods of data analysis, significant findings, and strengths and limitations [22]. This extraction was performed independently by two authors (AY & SS). The extracted results were compared, discrepancies were resolved through discussion, and tables were revised after consensus among all authors (AY, SS, AD, EM, CITG).

## Critical appraisal

The VAKS (Danish acronym for appraisal of qualitative studies) tool was used to critically appraise the articles [23]. This tool includes 30 items for critical appraisal of qualitative research. The items cover general requirements (6 items), credibility (7 items), transferability (5 items), dependability (6 items), and confirmability (6 items) of qualitative studies. The authors who extracted the data also performed the critical appraisal and scored each item on a four-point Likert scale from totally disagree = 1 to totally agree = 4. The total score for each criterion was calculated by adding the points for each criterion and dividing the number by the number of criteria. The cut-off values were strong>15, moderate = 10 to 15, and weak<10

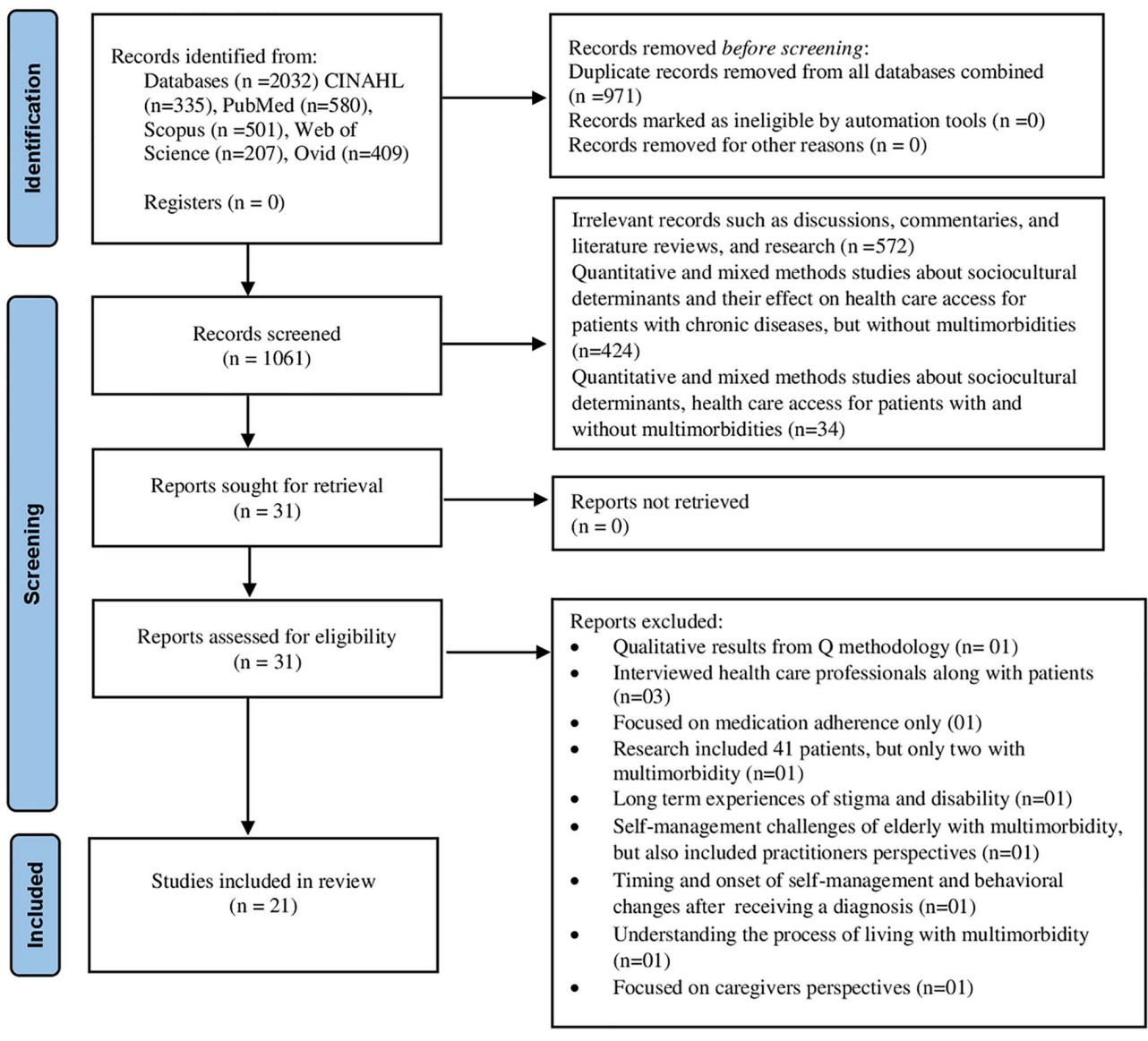

**Fig 1. Literature search.**

[23]. We performed this appraisal to evaluate the quality of the literature. However, we did not exclude any studies based on the quality scores. The studies were rated as strong, moderate, and weak. The weakly-rated studies were not excluded to prevent the risk of losing crucial contextual information [19]. However, during synthesis, the findings from strongly to moderately rated studies were given more weightage by sharing their findings and direct quotes under the developed themes. The findings from weakly rated studies were used to support the synthesized themes.

## Data synthesis

Thematic synthesis was used for data synthesis [24]. It includes coding, developing descriptive themes, and generating analytical themes. At the coding step, two reviewers (AY & SS)

independently reviewed the study findings, themes, sub-themes, categories, meanings units, participants' quotes, and the discussion section of the articles. Relevant phrases, words, findings, and quotes were coded using free line-by-line coding incorporating in vivo and open techniques. The similar codes were combined into descriptive themes based on their relations and underlying and apparent meanings and organized into hierarchical structures. It was ensured that descriptive themes were kept close to the original codes and findings [24]. The descriptive themes were evaluated by understanding the overall narrative of the studies and generating interpretations at an abstract level. The descriptive themes with similar and connected interpretations were synthesized into analytical themes (i.e., abstract themes generated after collating the descriptive themes and going beyond their apparent meanings). After the generation of analytical themes, the reviewers (AY & SS) conducted meetings to combine and finalize the themes. The remaining three reviewers (AD, EM, & CITG) reviewed the finalized themes and offered comments and suggestions for further improvement. During this synthesis process, reflective journals were maintained to write down interpretative notes and critical considerations for generating appropriate descriptions of findings. Reflective writing also enabled noting down pre-conceived biases and views about the topic and acknowledging how these assumptions may have influenced interpretations.

## Findings

### Literature overview and quality rating

Of 21 studies, most of the studies originated from the USA ($n = 5$) and Australia ($n = 4$), Canada ($n = 3$), UK ($n = 3$), and New Zealand ($n = 2$), followed by one each in South Africa, Ghana, the Netherlands, and Denmark. The commonly used research designs were descriptive qualitative approach ($n = 12$) and phenomenology (n = 6) while others used ethnography, case study, and secondary qualitative analysis. The sample size ranged from 11 to 85 participants with at least two multiple chronic conditions. The total number of participants in all the reviewed studies were 660 patients with multimorbidity and 27 formal or informal caregivers. The most common sampling techniques were purposive and convenience. The methods of data collection included in-depth and semi-structured interviews, document analysis, ethnographic and field observations, and focus groups. A wide range of data analysis methods were used, such as thematic analysis, conventional content analysis, Colaizzi's phenomenological method, and framework-based analysis (Table 1). Most studies were rated as high ($n = 8$) and moderate (n = 6) in quality. Four studies were rated weak in quality. The moderately and strongly rated studies had the following strengths: diverse and adequate samples, rigorous data analysis, detailed and thick description of methods and findings, triangulation, reflexivity, and data saturation (Table 1).

### Demographics

Based on the demographic data presented in all of the studies, it was identified altogether 252 men, 283 women, and one trans individual with an average of 4.5 multiple chronic diseases were interviewed. All of the individuals were above 30 years of age. Of these individuals, 219 were non-Caucasian (e.g., Asian, South Asian, African American, Latino, Hispanic), 208 were Caucasian, and 108 were Aboriginal. A large percentage (69%) of individuals belonged to the low and middle-income group, were retried, unemployed, or on a disability pension. The common chronic diseases included hypertension, diabetes, hyperlipidemia, cancer, coronary artery disease, chronic obstructive pulmonary disease, mental health problems, depression, generalized anxiety, arthritis, obesity, gastrointestinal problems, substance use disorder, and psoriasis.

**Table 1. Literature summary tables.**

| Authors/ Country | Purpose | Methods | Key Findings | Critical Appraisal |
|---|---|---|---|---|
| Bardach et al. [25] USA | To understand Appalachian residents' perspectives on multimorbidity management and prevention | **Design:** Descriptive qualitative **Theoretical framework:** NA **Setting:** Community and family medicine practice **Sample:** Patients above 50 years with an average of 4.7 multiple chronic conditions **Sample Size:** 41 **Sampling:** Purposive **Data collection:** In-depth interviews **Data analysis:** Conventional content analysis | Participants noted that a sense of isolation, inadequate community and financial resources, limited prevention awareness, and attitudinal factors affect their ability to attend the screening for colorectal cancer prevention. | **Strengths:** Thick description of methods and findings, adequate sample, robust data analysis, detailed information about study context, data saturation, and methods triangulation. **Limitations:** No member checking & discussion about reflexivity **Score:** 17.4 **Rating:** High |
| Bosire et al. [26] South Africa | To describe patients' experiences seeking care for comorbid HIV and diabetes | **Design:** Ethnography **Theoretical framework:** NA **Setting:** Tertiary hospital **Sample:** Patients above 30 years with at least two chronic conditions **Sample Size:** 15 **Sampling:** Purposive **Data collection:** In-depth narrative interviews & ethnographic visits **Data analysis:** Inductive thematic analysis | Fragmented care, multiple clinic appointments, conflicting information, and poor patient-provider communication impeded patients' access to care. Poverty, costly transport to the hospital, and food insecurity impeded the management of multimorbidities. | **Strengths:** Thick description of methods and findings, adequate sample, robust data analysis, detailed information about study context, methods triangulation, and reflexivity. **Limitations:** Small sample and no discussion about data saturation **Score:** 18.3 **Rating:** High |
| DiNapoli et al. [27] USA | To explore veterans' current disease-management practices, mental health treatment preferences, and challenges of living with multimorbidity | **Design:** Descriptive qualitative **Theoretical framework: NA** **Setting:** Primary care and/or behavioral health clinics **Sample:** Middle-aged and older veterans with at least two chronic conditions **Sample Size:** 34 **Sampling:** Purposive **Data collection:** Semi-structured interview and structured tools to assess the severity of disease **Data analysis:** Thematic analysis | Veterans noted medication, healthy lifestyle practices, and psychological stress management for self-management. However, they experienced barriers such as money, transportation, and stigma affecting self-care. | **Strengths:** Thick description of study findings, reasonable sample, robust method of data analysis. **Limitations:** No information about member checking, bracketing, audit trail, reflexivity, and no triangulation **Score:** 12.4 **Rating:** Moderate |
| Carusone et al. [28] Canada | To explore the hospital discharge and transition experience of complex patients over 6 weeks | **Design:** Case study **Theoretical framework:** NA **Setting:** Subacute care **Sample:** Aged 40 years or above, 5 male, 3 female, and one trans, with a mean of 5 chronic conditions **Sample Size:** 9 **Sampling:** Purposive **Data collection:** Interviews, review of discharge summaries, and chart abstraction **Data analysis:** Thematic analysis | Participants had a comprehensive discharge plan, but they felt overwhelmed following up with referrals, services, and medication adherence due to personal priorities and home environment. | **Strengths:** Thick description of methods and findings, adequate sample, robust data analysis, detailed information about study context, methods triangulation, audit trail, member checking, bracketing, and reflexivity. **Limitations:** Small sample and no discussion of data saturation **Score:** 18.4 **Rating:** High |

(*Continued*)

**Table 1.** (Continued)

| Authors/Country | Purpose | Methods | Key Findings | Critical Appraisal |
|---|---|---|---|---|
| Eton et al. [29] USA | To illustrate the burden of treatment from the perspective of the complex patient | **Design:** Descriptive qualitative **Theoretical framework:** Normalization process theory **Setting:** Medical outpatient pharmacist-led medication therapy management program **Sample:** Complex patients **Sample Size:** 32 **Sampling:** Purposive **Data collection: Data analysis:** Ritchie and Lewis' thematic analysis | Patients' burden of treatment included the necessary work for self-care for their health; problem-focused strategies. However, several personal, social, and system levels barriers affected their ability to engage in self-care. | **Strengths:** Thick description of study findings, reasonable sample, robust method of data analysis, researcher triangulation, and reflexivity **Limitations:** Limited information about the study context, no bracketing, no audit trail, or member checking **Score:** 14.8 **Rating:** Moderate |
| Lo et al. [30] Australia | To explore the perspectives of patients and their carers on factors influencing the health-care of those with co-morbid diabetes and chronic kidney disease. | **Design:** Descriptive qualitative **Theoretical framework:** Pragmatism **Setting:** Tertiary health care centers **Sample:** Patients and family caregivers **Sample Size:** 58 patients and 8 caregivers **Sampling:** Maximal variation sampling **Data collection:** Focus groups and semi-structured interviews **Data analysis:** Thematic analysis | Patients and their caregivers highlighted the significance of empowerment and self-management. Several Barriers to health care were noted such as poor access, poor continuity and coordination of care, and poor identification of psychological morbidity | **Strengths:** Thick description of study findings, reasonable sample, robust method of data analysis, audit trail, researcher triangulation, and data saturation **Limitations:** Limited information about the context and reflexivity **Score:** 17.8 **Rating:** High |
| Hardman et al. [31] Australia | To explore how do the specific demands of multimorbidity affect the burden and capacity of patients in rural settings | **Design:** Phenomenology **Theoretical framework:** Theory of Patient Capacity and Normalisation Process Theory **Setting:** Rural community health center **Sample:** Participants aged 47–72 years with 3–10 chronic conditions **Sample Size:** 13 **Sampling:** Purposive **Data collection:** Semi-structured interviews **Data analysis:** Framework based deductive analysis | Participants reported that Multimorbidity profoundly affected resource mobilization. The physical, psychological and financial capacities of participants were compromised. | **Strengths:** Robust data analysis and adequate sample **Limitations:** Limited information about study context and strategies to ensure rigor, no member checking, audit trail, no bracketing, no triangulation, and no discussion of reflexivity **Score:** 9.8 **Rating:** Weak |
| Ho et al. [32] Canada | To understand the challenges patients with multimorbidity face in accessing community care | **Design:** Secondary analysis **Theoretical framework:** NA **Setting:** Urban rehabilitation facility **Sample:** Patients with an average of five health conditions **Sample Size:** 116 **Sampling:** Purposive **Data collection:** Semi-structured interviews **Data analysis:** Exploratory interpretive analysis | Participants encountered health system level and at the individual patient level challenges such as scare resources, stress, incongruent care, access to health care, and financial strains. | **Strengths:** Thick description of study findings, reasonable sample, robust method of data analysis, audit trail, data saturation, researcher triangulation, and reflexivity **Limitations:** Secondary data analysis and limited information of the study context **Score:** 16.2 **Rating:** High |

(*Continued*)

**Table 1.** (Continued)

| Authors/ Country | Purpose | Methods | Key Findings | Critical Appraisal |
|---|---|---|---|---|
| McKinlay et al. [33] New Zealand | To explore the views of multimorbid patients about multimorbidity and available care | **Design:** Descriptive qualitative **Theoretical framework:** NA **Setting:** General care **Sample:** Samoan, Cambodian and Assyrian individuals with at least three chronic conditions **Sample Size:** 10 **Sampling:** Purposive **Data collection:** Focus groups and interviews **Data analysis:** Thematic analysis | Patients expressed confusion, information deficit, and limited understanding of chronic conditions. They also experienced resource constraints and financial issues. | **Strengths:** Methods triangulation **Limitations:** Superficial description of study methods and findings, limited information about study context, rigor, member checking, audit trail, bracketing, and reflexivity **Score:** 9.3 **Rating:** Weak |
| Corbett et al. [34] UK | To explore the experiences of older people with long-term chronic conditions about managing their health and meeting health-related goals after cancer treatment | **Design:** Descriptive qualitative **Theoretical framework:** NA **Setting:** Community **Sample:** At least 70 years old, with one or more chronic conditions and cancer **Sample Size:** 8 older adults and two caregivers (spouses) **Sampling:** Purposive **Data collection:** Interviews **Data analysis:** Framework analysis | Health care access issues, healthcare system issues, and relationships with care professionals affected the individuals' self-management abilities. | **Strengths:** Thick description of findings, adequate sample, robust data analysis, detailed information about study context **Limitations:** No member checking, no discussion about strategies to ensure rigor, no bracketing, no information about audit trail, and reflexivity **Score:** 9.8 **Rating:** Weak |
| Sav et al. [35] Australia | To explore treatment burden among people with various chronic conditions and comorbidities and unpaid carers. | **Design:** Descriptive qualitative **Theoretical framework:** Interpretive social paradigm **Setting:** Community **Sample:** Participants with at least two chronic conditions, only 10 patients had one chronic condition. Caregivers of participants **Sample Size:** 97 **Sampling:** Purposive and snowball **Data collection:** Interviews **Data analysis:** Iterative thematic analysis and constant comparative analysis | Patients experienced financial and travel burdens and received limited information from health care professionals about their treatment. The chronic illness affected the personal and social life of participants. | **Strengths:** Thick description of study findings, detailed information about study context, longitudinal assessment, reasonable sample, robust method of data analysis, and research triangulation **Limitations:** Limited information about data saturation and reflexivity **Score:** 14.8 **Rating:** Moderate |
| Ørtenblad et al. [36] Denmark | To explore the burden of treatment among people with multimorbidity by investigating the tension between everyday life and the health care system | **Design:** Ethnographic study **Theoretical framework:** NA **Setting:** Community **Sample:** Participants with at least three multiple chronic conditions **Sample Size:** 10 **Sampling: Purposive** **Data collection:** Interviews and participant observations **Data analysis:** Inductive analytical approach | Participants received inadequate support from health professionals in resolving their care-related dilemmas which included balancing family and social life, accommodating treatment and work, and clashes with health care professionals | **Strengths:** Thick description of study findings, detailed information about study context, longitudinal assessment, reasonable sample, robust method of data analysis, audit trail, and triangulation **Limitations:** Limited information about data saturation and reflexivity **Score:** 14.8 **Rating:** Moderate |

*(Continued)*

**Table 1.** (Continued)

| Authors/ Country | Purpose | Methods | Key Findings | Critical Appraisal |
|---|---|---|---|---|
| Morgan et al. [37] Ghana | To explore the perceptions and experiences of women living with multi-morbidity | **Design:** Descriptive qualitative **Theoretical framework:** Cumulative complexity model **Setting:** Polyclinics **Sample:** Women with at least two multiple chronic conditions **Sample Size:** 20 **Sampling:** Stratified purposive sampling **Data collection:** In-depth interviews **Data analysis:** Thematic analysis | Spirituality and disease-related stigmatization impacted health experience. Women depended on family and community to provide financial support for care and treatment. | **Strengths:** Thick description of study findings, detailed information about study context, reasonable sample, robust method of data analysis, audit trail, data saturation, researcher triangulation, and reflexivity **Limitations:** No discussion of bracketing **Score**: 18.4 **Rating**: High |
| Signal et al. [38] New Zealand | To understand patients' perspectives of living with multimorbidity. | **Design:** Phenomenology **Theoretical framework:** NA **Setting:** Primary healthcare **Sample:** Participants with at least four chronic conditions **Sample Size:** 61 **Sampling:** Purposive **Data collection:** Focus groups and interviews **Data analysis:** Thematic analysis | Participants noted challenges concerning coping with disease management and dealing with health system-related issues such as access and care. | **Strengths:** Thick description of study findings, detailed information about study context, reasonable sample, and method triangulation **Limitations:** Inappropriate method of data analysis, unclear if hermeneutic or descriptive phenomenology, no phenomenological bracketing, reflexivity, member checking, or audit trail. **Score:** 9.8 **Rating:** Weak |
| Duguay et al. [39] Canada | To explore adults' experiences of multimorbidity | **Design:** Descriptive phenomenology **Theoretical framework:** NA **Setting:** Primary care **Sample:** Patient with multimorbidities **Sample Size:** 11 **Sampling:** Purposive **Data collection:** Two Semi-structured interviews with each participant **Data analysis:** Colaizzi's method of analysis | Managing multimorbidity was a complex endeavor and participants experienced many social and personal issues affecting their self-care and access to health care. | **Strengths:** Thick description of study findings, reasonable sample, robust method of data analysis, phenomenological bracketing, researcher triangulation, reflexivity, and audit trail **Limitations:** Limited information about the study context and data saturation **Score:** 18.4 **Rating:** High |
| Maneze et al. [40] Australia | To explore the diabetic patients' experience, barriers, and facilitators of multidisciplinary care | **Design:** Descriptive qualitative **Theoretical framework:** NA **Setting:** Local district hospital **Sample:** Patients with diabetes **Sample Size:** 13 **Sampling:** Convenience **Data collection:** Semi-structured interviews **Data analysis:** Thematic analysis | Patients encountered multiple physical and psychosocial barriers such as lack of a dedicated coordinator of care, follow-up and support, financial difficulties, lack of transport, and language barriers. | **Strengths:** Reasonable sample and thick description of study findings **Limitations:** Limited information about study context, rigor, member checking, audit trail, bracketing, saturation, triangulation, and reflexivity **Score:** 9.8 **Rating:** Weak |

(*Continued*)

**Table 1.** (Continued)

| Authors/ Country | Purpose | Methods | Key Findings | Critical Appraisal |
|---|---|---|---|---|
| Mason et al. [41] UK | To explore experiences and perceptions of people with advanced multimorbidity | **Design:** Descriptive qualitative **Theoretical framework:** NA **Setting:** General practice & multiethnic urban and rural locations **Sample:** Patients with multimorbidity and their family members **Sample Size:** 87 interviews with 37 patients and 17 carers **Sampling:** Purposive **Data collection:** Interviews **Data analysis:** Constructivist thematic analysis | Individuals restricted their interactions with care professionals to preserve autonomy. They also encounter difficulty in understanding their multiple conditions, accessing medications, and available services and support. | **Strengths:** Reasonable sample, data saturation, and method triangulation **Limitations:** Superficial description of study methods and findings, limited information about study context, rigor, member checking, audit trail, bracketing, and reflexivity **Score:** 10.6 **Rating:** Moderate |
| Villena & Chesla [42] USA | To understand the social and structural barriers that individuals with co-occurring disorders encounter in regard to their health care | **Design:** Interpretive hermeneutic **Theoretical framework:** NA **Setting:** Community centers **Sample:** Participants with at least two co-occurring disorders **Sample Size:** 20 **Sampling:** Purposive **Data collection:** Interviews **Data analysis:** Benner's interpretative analysis | Participants perceived Social and structural barriers included difficult interpersonal relationships with health care professionals, negotiating an arduous health care system, and trying to manage health conditions while living in an unstable shelter. | **Strengths:** Thick description of study findings, reasonable sample, robust method of data analysis, member checking, reflexivity, bracketing, and researcher triangulation **Limitations:** Limited information about study context **Score:** 18.4 **Rating:** High |
| van Merode et al. [43] The Netherlands | To explore experiences of individuals with multimorbidity regarding daily symptom burden and its management | **Design:** Descriptive qualitative **Theoretical framework:** NA **Setting:** Primary care **Sample:** Seven men and 15 women with at least two or more conditions **Sample Size:** 22 **Sampling:** Purposive sampling **Data collection:** In-depth interviews **Data analysis:** Thematic analysis | Care coordination, health care system access and services related, scarce resources, communication issues with care professionals, and medical related issues affected individuals' self-management. | **Strengths:** Detailed information of context, adequate sample size, data saturation **Limitations:** No discussion of bracketing, limited information about the method of data analysis, audit trail, no triangulation, and no discussion of reflexivity or strategies to ensure rigor **Score:** 9.8 **Rating:** Weak |
| Porter et al. [44] UK | To explore experiences of individuals with multimorbidity | **Design:** Descriptive qualitative longitudinal **Theoretical framework:** Phenomenology **Setting:** Community **Sample:** Eight women and seven men with at least two multiple chronic conditions **Sample Size:** 15 individuals, 27 interviews **Sampling:** Purposive sampling **Data collection:** In-depth interviews **Data analysis:** Constructivist Grounded Theory based analysis | Individuals challenged the concept of illness and provided their own accounts of normality. Interactions and relationship with health care professionals informed their experiences and management. | **Strengths:** Thick description of the context and adequate sample size **Limitations:** No discussion of bracketing, mismatch of methodology and analytical method, no audit trail, triangulation, and reflexivity **Score:** 8.8 **Rating:** Weak |

(*Continued*)

**Table 1.** (Continued)

| Authors/ Country | Purpose | Methods | Key Findings | Critical Appraisal |
|---|---|---|---|---|
| Corser et al. [45] USA | To explore personal self-management perspectives heavily comorbid primary care adults with at least four chronic health conditions | **Design:** Descriptive qualitative **Theoretical framework:** NA **Setting:** Family medicine clinic **Sample:** At least 21 years old, 14 men and four women with at least two chronic conditions **Sample Size:** 18 **Sampling:** Purposive **Data collection:** Focus groups and chart audits **Data analysis:** Content analysis | Participants experienced physical and psychological limitations during self-management. Access to health care was impeded by poverty and the limitations of the health care system. | **Strengths:** Thick description of methods, adequate sample, robust data analysis, detailed information about study context, methods triangulation **Limitations:** No member checking, no bracketing, no information about audit trail, and reflexivity **Score:** 13.6 **Rating:** Moderate |

## Thematic synthesis

Eight descriptive themes and three analytical themes were developed from all studies [25–45]. The themes are discussed and the detailed thematic synthesis is illustrated in Table 2.

### Analytical theme one: Contextualized personal and structural vulnerabilities impact access to care and navigation of health care delivery systems

This analytical theme comprised three descriptive themes: living circumstances and vulnerabilities, scarce health resources, and intrusive health care delivery system. Individuals with multimorbidities in self-management were constrained by personal and structural vulnerabilities that varied across different contexts and life circumstances. Limited health care facilities and personal and community resources impacted their access to distant health care services. If they could access the services, they would experience challenges navigating the arduous health care system. This theme was more apparent in the narratives of individuals with diabetes, hypertension, respiratory problems, substance abuse, and mental health problems.

**Living circumstances and vulnerabilities.** Individuals with multimorbidities' life circumstances change with ongoing personal and structural issues resulting in continuing vulnerabilities. The individuals describe financial issues, low income, rural living conditions, unemployment, transportation issues, poor housing conditions, and physical disabilities affecting their self-management of multimorbidities [25–35]. These circumstances made them vulnerable to poor and inadequate self-management.

Many individuals described that initially, they had the finances to manage their health care problems. However, as their health care conditions deteriorated and needed more self-care, they ran out of funds. They had to pay out of pocket for basic medical needs such as medications and equipment for self-care. Lo et al. [30] reported that individuals felt stressed because they had to travel a long distance to seek health services, payout of pocket for essential services, and incidental costs such as car parking. Limited finances made it difficult for them to make a difficult choice of staying home and avoiding seeking care.

Such living circumstances prevented them from seeking care and engaging in self-management and affected their emotional health, leading to stress, confusion, uncertainties, and fears. Sav et al. [35] noted that medicinal and consultation costs instigated the financial burden. The individuals felt worried and concerned if they could manage their multiple health issues. Analysis of individuals' experiences in the reviewed studies indicated that Self-management was for

**Table 2.  Thematic synthesis.**

| Codes | Descriptive Themes | Analytical Theme |
|---|---|---|
| Low income [25, 26, 28, 42] | Living circumstances and vulnerabilities | Contextualized personal and structural vulnerabilities impact access to care and navigation of health care delivery systems |
| Unemployment [25–27, 33] | | |
| Financial constraints [25–27, 29–34, 42–45] | | |
| Limited Transportation [25–27, 30, 32, 34, 35, 38, 43–45] | | |
| Costly transportation [25, 27, 35, 38] | | |
| Congested homes many people in a single room or a house [26, 42] | | |
| Out of pocket cost for medications [25, 29, 30, 32, 34] | | |
| Poor housing [26, 27, 38, 42] | | |
| Rural living [26, 32, 37, 38, 42] | | |
| Living alone [29, 38, 42] | | |
| Limited education/literacy level [25, 36] | | |
| Disability prevented from traveling to health care facilities [30, 43] | | |
| Non-sustainable private health insurances [31, 32] | Scarce health care resources | |
| Cutbacks and reductions in insurances [32] | | |
| Inadequate medical facilities in the community [25, 27, 28, 30, 32, 35] | | |
| Ineligibility to basic care and services [27, 32–34] | | |
| No health insurance [25, 36] | | |
| Distant tertiary hospitals and specialized care facilities [26, 30] | | |
| No family doctors [32, 37] | | |
| Lack of availability of language interpreters [30] | | |
| Emphasis on using traditional and herbal medicine [26, 36] | | |
| No trust in the health care system [27, 32, 38, 42] | Intrusive health care delivery system | |
| Bureaucracy in the health care system [29, 36, 37, 42] | | |
| Switching health care professionals [32, 34, 42] | | |
| Difficulty in getting paperwork to avail health care [26, 28, 29, 31, 32] | | |
| Difficulty scheduling and changing appointments [27, 30, 32, 34–36, 38] | | |
| Contradictory and disjointed formal care delivery [29, 30, 32, 34, 37, 42, 43] | | |
| Long waiting times [32, 35, 37] | | |
| Lack of coordination among health care professionals when planning care [30, 34, 36, 39, 40, 42–45] | | |
| Ethnic discrimination [26] | Intercultural issues | Social and family struggles limit health care access and at home self-care |
| Non-white cultural background impacts care access and resources [30] | | |
| Women face more issues in accessing care and engaging in self-care [36] | | |
| Lack of family support [39, 40] | Intrapersonal familial struggles | |
| Family as the source of stress [28, 36, 39] | | |
| Competing priorities e.g., caring for family members [28] | | |
| Engaging in self-care evokes feelings of guilt with family and friends [29] | | |
| Dependent on family members for finances and resources [31, 36, 37] | | |
| Sacrificing self-care [29] | | |

*(Continued)*

**Table 2.** (Continued)

| Codes | Descriptive Themes | Analytical Theme |
|---|---|---|
| Stigmatized health problems [27, 36, 38] | Epistemic Injustice influencing motivation to access care | Fragmented interpersonal relationship reduces self-management preparedness and ability |
| Felt voiceless in front of health care professionals [29, 34, 36, 42] | | |
| Felt misunderstood [40, 42] | | |
| Health care professionals ignore holistic needs and care [36, 40] | | |
| Struggled to explain their perspective [42] | | |
| Limited trust in health care professionals [27, 29, 42] | Relational conflicts | |
| Poor communication about disease, care, and tests from health care professionals [29, 30, 34, 39, 43–45] | | |
| Conflicts with health care professionals [29, 30, 34, 35] | | |
| Lack of education from health care professionals [30, 33, 34, 39, 40, 42] | Substandard health education | |
| Jargon loaded health education [30, 33] | | |
| Failure of health care professionals to provide sufficient advice [27, 31, 33, 34, 39, 40] | | |
| Trial and error learning for self-education about chronic conditions [31] | | |
| Lack of follow-up after health education [34, 39, 40] | | |

the privileged, and individuals with structural, personal, and social vulnerabilities struggled to care for themselves while also striving to meet the needs of their families.

> "The money is not enough. I can't even afford my rent and diabetes foods. I also need to buy foods and medication for my wife. I only try to buy whatever I can, but I never get to achieve what the doctor recommends [. . .]. Although I adhere to my medications, I do not see much changes. My diabetes and blood pressure seem to be getting out of control

[26, p. 379].

**Scarce health care resources.** Individuals described a lack of health care resources and services which made self-management a complex and tiresome process. The health care centers, and specialized care services were at a distance from individuals' homes and most of them did not have transportation. In the communities, there was a shortage of medical supplies [25, 27, 28, 30, 32, 34, 36]. The individuals did not have health care insurance to meet the expenses of medical and specialized care. Those who had insurance received reductions and cutbacks because of their multiple complex health care needs and they expressed that their insurance was not sustainable. Ho et al. [32] noted that some individuals received home care services and covered the expenses through private insurance. However, financial challenges and disease burden resulted in cutbacks in the hours and visits received from formal caregivers.

Lack of health care resources in the communities and the inability to travel to distant hospitals prompted many individuals to put greater emphasis on home remedies and herbal medicine to manage their health issues [26, 34] The use of herbal and traditional medicine was also prompted by social and cultural beliefs, lack of medical centers, and peer pressure. Given the shortage of medical facilities, many traditional healers often encouraged individuals to give up their prescribed medicines. Morgan et al. [37] described that women with HIV and multimorbidities bought herbal medicines from local vendors and churches.

Many individuals did not have family doctors so it was almost impossible for them to seek medical advice and consultation when needed [32] The lack of facilities and health centers in the communities aggravated their conditions, further adding to the issues of seeking medical support. These individuals had to travel to far-off places to talk to specialists. Signal et al. [38] discussed that many individuals had to see professionals in primary care because they no longer had a family doctor. Individuals with diverse ethnic backgrounds noted that when they can access health care services, they cannot explain their issues because of the lack of interpreters in health care centers [30]

"They get cut back. Like with the PSW, I was given 3 times a week and then when they start getting towards the end of the budget, the money's running down, then you get cut back. And sometimes you get it back, sometimes you don't. . .."

[32, p. 1314].

**Intrusive health care delivery system.**   Individuals discussed that the health care system is challenging to navigate because the bureaucracy gets in their face when they try to seek care and support. They explained the difficulties in scheduling their appointments, switching health care professionals, and providing the paperwork for seeking health care services [29, 30, 32, 34, 36–39]. They noted a lack of coordination among health care professionals resulting in costing them their appointments and consultations [30, 35, 38, 40–42]. Individuals living in poor housing and unstable shelters experienced struggles to seek medical attention when needed. They experienced logistic struggles in making, attending, and changing their appointments [32, 35, 42].

Individuals noted bureaucracies that affected their ability to seek health care services. Some individuals did not have the paperwork, or the documents needed to avail themselves of the services. While those who had the paperwork, the government, and the institutions did not approve it, or they had to wait a long time before it was approved [26, 28–32]. Villena and Chelsa [42] noted that navigating institutional bureaucracies within the Medicaid system was a consistent struggle. This struggle was worsened if they also lived in unkempt and dangerous housing accommodations. Such health system issues combined with social vulnerabilities affected individuals' trust in the health system and its efficiency to address their multiple needs [27, 32, 39].

"The State said I'm not eligible for Medicaid. But we have a diagnosis of diabetes, emphysema, and now, cancer. Why? Because I worked? I gave up my job, and I still didn't get the Medicaid. I had no medical insurance. I had to fight them. I went to Legal Aid. The State said I defrauded them because I worked. I finally got my Medicaid in July. I got the energy and got up and fought the system and their red tape because I needed to. I needed the Medicaid, or what was I gonna do? Just die? As it stands, I owe $182,000 worth of medical bills"

[42, p. 82].

## Analytical theme two: Social and family struggles limit health care access and at home self-care

Individuals expressed that some covert sociocultural issues about social hierarchies and family systems impact their ability to access health care services and perform self-care at

home. Individuals with different ethnicities were often denied essential health care services because of bureaucratic issues. While women had to depend on their families to access health care and financial expenses for self-care. These two struggles were widespread for individuals in low and middle-income countries and those with different ethnic backgrounds. This theme was more apparent in the narratives of individuals with diabetes, hypertension, coronary artery disease, AIDS, chronic obstructive pulmonary disease, and mental health problems.

**Intercultural issues.**   Two intercultural struggles were ethnic and gender differences. Individuals from diverse ethnic backgrounds experienced more challenges to access healthcare services than native residents. The bureaucratic health care system greatly influenced this intercultural issue because those with different ethnic backgrounds had to provide more paperwork to prove that they qualified for health care services. Bosire et al. [26] shared an account of a woman who did not speak and understand English and required her sister's help during- clinical appointments. Her mother was Lesotho and her father was a South African. They passed away because of HIV. She and her siblings were denied the right to be a South African citizen. Therefore, she did not have any proof making her eligible to access primary care services offered by the government. Lo et al. [30] discussed that some patients expressed that being a non-English speaker with a different ethnicity and culture characterized by apathy negatively influenced self-management and dietary restrictions. Pertaining to gender discrimination and self-management, only one study noted that [37] women face more issues in accessing care and engaging in self-care because they were dependent on their families and community to offset the financial burden of chronic disease management and treatment.

> "As for that one my child, eating is even difficult for us. . . .The time that I am supposed to eat, I don't get food to eat . . . It is only the boys who support me a little. It is almost one year since I got paralysed and none of them have called on phone to find out how I am doing. If I don't call them, then I would not hear from them. They are all into businesses, but they do not mind me, let alone to support me financially"

> [37, p.11]

**Intrapersonal familial struggles.**   The intrapersonal familial struggle was conceptualized as individuals' tensions within their families and, or with family members, affecting their self-management. Individuals discussed a lack of family support [41, 42] Many of these individuals had to rely on their families for financial, physical, and emotional support. However, often the support was not available [31, 37, 38]. Some individuals considered their family as the source of stress which prevented them from fully engaging in self-management of their multiple health care issues [28, 37, 40]. Individuals also noted that focusing on self-care sometimes led to feelings of guilt for not caring for their families [29, 40]. They often had to make a deliberate choice of competing family priorities [28] and sacrifice their own needs for their family members which created intrapersonal tensions [29].

> "I mean a lot of that has been completely on my wife as far as paying the house note and most of the bills through a lot of this, and that's caused tension, of course. . . We have been on this kind of rocky road"

> [29, p. 44].

### Analytical theme three: Fragmented interpersonal relationship reduces self-management preparedness and ability

This analytical theme captured the patient-care professional relationship and how it affected the self-management ability of individuals with multiple chronic conditions. It entails three descriptive themes illustrating epistemic battles among patients and health care professionals, their relational conflicts, and the quality of health education provided by health care professionals. The interaction between providers and patients was fragmented and negatively influenced individuals' self-management ability and preparedness. The individuals had to mainly rely on themselves to perform adequate self-care. This theme was more apparent in the narratives of individuals with diabetes, coronary artery disease, chronic obstructive pulmonary disease, respiratory problems, psoriasis, and menta health problems.

**Epistemic injustice influencing motivation to access care.** Many individuals had health care problems (e.g., AIDS), which are often stigmatized among the general public and communities. Others experienced stigmatization because of their age, gender, and living conditions [27, 37–39]. The individuals noted that they felt voiceless in front of health care professionals because their concerns and complaints were not listened to attentively [29, 34, 37, 42]. Often the health care professionals asked the patients not to schedule any appointments unless necessary without considering the needs and viewpoints of the patients [41, 43]. Some individuals felt misunderstood because they often did not know the typical labels and descriptive language to describe their health problems to health care professionals [41–43]. Health care professionals ignore holistic needs and care and tend to address one disease at a time [34, 37, 41, 42]. Individuals were concerned that addressing one disease at one time did not offer them adequate knowledge to manage any interrelated symptoms of multiple diseases [34, 41, 42]. Generally, the individuals struggled to explain their perspectives to their health care professionals [42].

> "You can phone in and the doctor will talk to you or you can phone a nurse and they will talk to you. 'Please don't make an appointment unless it is absolutely necessary.' Well, what do you know is absolutely necessary? I have been rushed into hospital because I have left it too long, because I won't bother them"

[41, p. 62].

**Relational conflicts.** Individuals discussed that while navigating the complex health care services and managing their chronic issues, they had many conflicts with health care professionals [29, 30, 36, 42]. The conflicts were varied in nature. For example, clashes among patients and clinicians regarding patient preferences and professionals' viewpoints about treatment modalities and goals [36], patients' struggling to convince health care professionals of the urgency of their health care needs due to deliberate care neglect [42], and conflicts about poor and wrong communication about disease, care, and tests from health care professionals [29–32]. Many individuals noted that the source of the conflicts was often non-caring attitudes, lack of attention from health care professionals, and bureaucracies surrounding health care access and services [26, 29, 30, 41]. Relational conflicts shattered patients' trust in health care professionals, thereby affecting individuals' ability and preparedness to perform self-management [27, 29, 42]. The lack of trust in health care professionals also stemmed from past negative experiences in health care [27].

> "A few days after my appointment, I noticed that my sugars were high in the morning when they shouldn't have been. So, then that's when I realized, oh, this is N, and not 70/30.

I called and left a message on the doctor's voice mail. She was on vacation, but I knew that she would pick it up Monday. And she called on Monday and said, "Oh, yes, I did give you the wrong insulin." She tried to correct her mistake by telling me to go back to the clinic that day and get another prescription"

[42, p. 80].

**Substandard health education.** Individuals raised concerns about substandard health education from the health care professionals. Many individuals noted that the health care professionals completely ignored educating them about their disease, treatment, and care modalities [30, 33–35, 41, 42, 44, 45]. However, others noted that the education was delivered with jargon-loaded language without considering the education level and needs of the patients [30, 33]. The self-management of individuals was greatly affected by the failure of health care professionals to provide sufficient advice on managing complications and urgent health care issues interlined to multiple diseases [27, 31, 34, 35, 41–44]. Concerns were raised that health professionals were time-poor and did not take the initiative to offer any health education or explanations unless asked by the patients [30, 34, 43]. If health education was offered, individuals noted a lack of follow-up from health care professionals [38, 40, 41]. Given the lack of quality health education, most patients relied on trial and error learning for self-education and explored their resources to learn about disease management [31].

"...and I ask the doctor, 'What's diabetes?' Sugar is high . . . even today I still puzzled because they are not explained properly how high . . . the sugar [is]. All they say is 'diabetes'

[33, p. 231].

## Discussion

This review aimed to synthesize literature about experiences and perspectives of individuals with multimorbidities to understand sociocultural and patient and health care professional level factors affecting their self-management ability. The principal findings of this review are that individuals with multimorbidities self-management are affected by personal and social vulnerabilities, familial struggles, and fragmented interpersonal relationships with health care professionals. Sociocultural factors such as poverty, economic circumstances, living and housing conditions greatly affect their access to and usage of health care services to complement their own self-management efforts. If they are able to access health care services and resources, they struggle to navigate the arduous health care system and bureaucracies and experience epistemic struggles and conflicts with health care professionals. These findings are consistent with previous reviews [3, 6] that identified that individuals with multimorbidities struggle to access health care and fail to receive holistic care for their health care issues.

The findings of this review bring into attention the importance of structural competency for health care professionals and the need to design better structural interventions in health care organizations. Structural competency enables in understanding the impact of sociocultural and demographic determinants such as race, class, gender, and ethnicity on health care professionals' encounters in relation to the broader structural contexts in which their encounters take place [46]. Health care professionals should be trained to become more culturally competent in order to address sociocultural factors and foster the relationship among health care professionals and individuals with multimorbidities. Additionally, training should also

include effective management of chronic disease management and care processes to better prepare individuals for self-management [47].

Health care organizations should focus on developing and implementing more targeted interventions to provide effective care to individuals with multimorbidities. Dineen-Griffin and colleagues [14], based on a systematic review of 58 studies, identified several interventions to promote self-management among individuals with multimorbidity. The commonly reported interventions included: knowledge and training programs, development of personalized self-care actions pans, coping and stress management programs, and medication adherence and lifestyle improvement programs. These interventions can be tailored and adapted after considering the sociocultural and patient-professional factors identified in the review, and placing mechanisms to address the negative impact of sociocultural determinants. The possible mechanisms could include greater involvement of family and informal caregivers, use of virtual and telephonic care in urgent times, involvement of multidisciplinary teams in patient care, and greater emphasis on community and home-based care with health care professionals enhanced cultural and structural competencies [48, 49]. The utilization of such mechanisms is particularly important in low and middle-income countries, which may have a public-funded health care system but no health insurance for meeting additional expenses. The effectiveness of such comprehensive care programs and interventions has been established across contexts [48–52]. Therefore, existing interventions implemented in diverse contexts could be contextualized using implementation science methods and tools to implement in new settings. One critical consideration for designing and contextualizing such integrated and comprehensive care programs is that these programs should be person-centered and tailored to the felt needs of individuals with multimorbidities [50, 52]. Person-centered programs should include better assessment of patients' needs, therapeutic relationships among health care professionals and patients, and sustainable health education to foster self-management [47, 50].

## Limitations and implications for future research

Only English language studies were included which could have resulted in missing any additional studies on the very topic. While performing thematic synthesis subjective and value judgments were made for developing descriptive and analytical themes which may have influenced the validity of the review. Nevertheless, multiple reviews and independent assessments were taken to ensure that personal biases do not affect the interpretation of reviewed literature. Most of the reviewed studies originated from the USA, Canada, and Australia. Therefore, the transferability of findings to other contexts may be limited. Further research is warranted to explore sociocultural and patient-professional factors affecting self-management of individuals with multimorbidity in low- and middle-income countries. We conducted a meta-synthesis of qualitative studies which, by its nature, does not offer statistical analyses of sociocultural factors affecting self-management. Therefore, it would be useful to conduct future systematic review of quantitative literature to determine the associations among various sociocultural factors and self-management.

## Conclusions

Self-management is instrumental for individuals with multimorbidities in managing their complex health care issues and unanticipated potential complications. However, engagement in self-management for these individuals is hampered by under-resourced community and health care environments, personal and structural vulnerabilities, familial and interpersonal conflicts, and disjointed and fragmented relationships with health care professionals. There is

a dire need to ensure optimal community resources to support individuals to address and navigate complexities associated with accessing care and effectively managing their illnesses. The relationship between health care professionals and individuals with multimorbidities was found to be fragmented due to epistemic struggles and relational conflicts. Therefore, training for structural and cultural competence of health care professionals can be promising to better prepare them to address sociocultural determinants and negative individual level factors affecting their abilities to provide effective and person-centered care.

## Supporting information

**S1 Checklist. PRISMA 2020 checklist.**
(DOCX)

**S1 Table. Search strategy for PubMed.**
(DOCX)

## Author Contributions

**Conceptualization:** Ahtisham Younas, Sharoon Shahzad, Angela Durante.

**Data curation:** Sharoon Shahzad.

**Formal analysis:** Ahtisham Younas, Sharoon Shahzad, Clara Isabel Tejada-Garrido, Angela Durante.

**Methodology:** Ahtisham Younas, Sharoon Shahzad, Clara Isabel Tejada-Garrido, Esther Nyangate Monari, Angela Durante.

**Validation:** Ahtisham Younas, Clara Isabel Tejada-Garrido, Esther Nyangate Monari, Angela Durante.

**Writing – original draft:** Ahtisham Younas, Sharoon Shahzad, Clara Isabel Tejada-Garrido, Esther Nyangate Monari, Angela Durante.

**Writing – review & editing:** Ahtisham Younas, Sharoon Shahzad, Clara Isabel Tejada-Garrido, Esther Nyangate Monari, Angela Durante.

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
