## [Decision Letter · Decision Letter 0]

11 Jul 2023

PGPH-D-23-01010

Sociocultural and Patient-Health Care Professional Related Factors Influencing Self-Management of Patients with Multimorbidities: A Thematic Synthesis

Dear Dr. Younas,

Thank you for submitting your manuscript to PLOS Global Public Health. After careful consideration, we feel that it has merit but does not fully meet PLOS Global Public Health’s publication criteria as it currently stands. Therefore, we invite you to submit a revised version of the manuscript that addresses the points raised during the review process.

We look forward to receiving your revised manuscript.

Kind regards,

Mathew Sunil George

Academic Editor

Journal Requirements:

2. Please send a completed 'Competing Interests' statement, including any COIs declared by your co-authors. If you have no competing interests to declare, please state "The authors have declared that no competing interests exist". Otherwise please declare all competing interests beginning with twhe statement "I have read the journal's policy and the authors of this manuscript have the following competing interests:"

3. Please provide separate figure files in .tif or .eps format only and remove any figures embedded in your manuscript file. Please also ensure all files are under our size limit of 10MB.

Additional Editor Comments (if provided):

Reviewers' comments:

Reviewer's Responses to Questions

**Comments to the Author**

1. Does this manuscript meet PLOS Global Public Health’s publication criteria? Is the manuscript technically sound, and do the data support the conclusions? The manuscript must describe methodologically and ethically rigorous research with conclusions that are appropriately drawn based on the data presented.

Reviewer #1: Yes

Reviewer #2: Partly

2. Has the statistical analysis been performed appropriately and rigorously?

Reviewer #1: Yes

Reviewer #2: N/A

3. Have the authors made all data underlying the findings in their manuscript fully available (please refer to the Data Availability Statement at the start of the manuscript PDF file)?

Reviewer #1: Yes

Reviewer #2: Yes

4. Is the manuscript presented in an intelligible fashion and written in standard English?

Reviewer #1: Yes

Reviewer #2: Yes

5. Review Comments to the Author

Reviewer #1: Overall, the study appears to be well-structured and contributes valuable insights into the sociocultural and patient-health care professional factors affecting self-management in patients with multimorbidities. However, to further strengthen the study, the following comments from my side:

The information on gender distribution, age range, and racial/ethnic backgrounds adds to the understanding of the population under study. However, it is suggested to mention the total number of participants included in the meta-synthesis, as this would provide a clearer perspective on the sample size and generalizability of the findings.

While the study highlights the need for targeted interventions to address the identified barriers, it should include a discussion or analysis of existing interventions that have been implemented in diverse contexts.

Inclusion of more literature having statistical analyses, such as regression models or correlation analyses, could help determine the associations between sociocultural factors (e.g., socioeconomic status, cultural beliefs, social support) and self-management outcomes among individuals with multimorbidities.

The most of the reviewed studies originated from from the USA (n=5) and Australia (n=4), Canada (n=3), UK (n=3), and New Zealand (n=2), followed by one each in South Africa, Ghana, the Netherlands, and Denmark, which raises concerns about the generalizability of the findings to other geographical regions and cultural contexts. Conducting a more extensive literature review to include studies from diverse regions and populations could enhance the study's generalizability and provide a more comprehensive understanding of the sociocultural and patient-health care professional factors influencing self-management among individuals with multimorbidities.

Reviewer #2: Thanks for taking the initiative to conduct the meta-synthesis of multimorbidities patients and showing how sociocultural and patient-health care professional-related factors influence self-management, which is a very important issue. However, you can consider the following issues to make the paper improve.

- Most of the findings indicate that it is mostly focused on the experience of different ethnic people living in poverty, less educated, and high-income countries which should be cleared in the method section and somehow in the title.

- The paper could be enriched if the authors have given more examples from low-middle-income countries.

- Since the authors have discussed the measurement to minimization of the problems, I think it would be better to discuss the problems according to the context/high-income/low-income country.

- There are lots of headings (excluding findings) in the paper which is sometimes confusing and I don’t think all of them are necessary

- Literature Overview and Quality Rating and Demographics – these two paragraphs should go to the Method section rather than Findings.

- A few points are in the PDF version

- Please check the highlighted words or lines – I didn’t understand those well

6. PLOS authors have the option to publish the peer review history of their article (what does this mean?). If published, this will include your full peer review and any attached files.

**Do you want your identity to be public for this peer review?** For information about this choice, including consent withdrawal, please see our Privacy Policy.

Reviewer #1: **Yes: **Dr Md Abul Hasan

Reviewer #2: No

---

## [Decision Letter · Decision Letter 1]

25 Aug 2023

Sociocultural and Patient-Health Care Professional Related Factors Influencing Self-Management of Multiethnic Patients with Multimorbidities: A Thematic Synthesis

PGPH-D-23-01010R1

Dear Dr Younas, 

We are pleased to inform you that your manuscript 'Sociocultural and Patient-Health Care Professional Related Factors Influencing Self-Management of Multiethnic Patients with Multimorbidities: A Thematic Synthesis' has been provisionally accepted for publication in PLOS Global Public Health.

Best regards,

Mathew Sunil George

Academic Editor

Reviewer Comments (if any, and for reference):

Reviewer's Responses to Questions

**Comments to the Author**

1. If the authors have adequately addressed your comments raised in a previous round of review and you feel that this manuscript is now acceptable for publication, you may indicate that here to bypass the “Comments to the Author” section, enter your conflict of interest statement in the “Confidential to Editor” section, and submit your "Accept" recommendation.

Reviewer #1: All comments have been addressed

Reviewer #2: (No Response)

2. Does this manuscript meet PLOS Global Public Health’s publication criteria? Is the manuscript technically sound, and do the data support the conclusions? The manuscript must describe methodologically and ethically rigorous research with conclusions that are appropriately drawn based on the data presented.

Reviewer #1: Yes

Reviewer #2: Yes

3. Has the statistical analysis been performed appropriately and rigorously?

Reviewer #1: Yes

Reviewer #2: N/A

4. Have the authors made all data underlying the findings in their manuscript fully available (please refer to the Data Availability Statement at the start of the manuscript PDF file)?

Reviewer #1: Yes

Reviewer #2: Yes

5. Is the manuscript presented in an intelligible fashion and written in standard English?

Reviewer #1: Yes

Reviewer #2: No

6. Review Comments to the Author

Reviewer #1: The manuscript now appears well-structured, addressing key comments and suggestions. It offers valuable insights into the sociocultural and patient-healthcare professional factors impacting self-management among patients with multimorbidities.

Reviewer #2: Many thanks for addressing the points raised by reviewers. Authors should review again to identify typographical errors. I found one in Pg 21 (first paragraph) and it might be plans not pans.

7. PLOS authors have the option to publish the peer review history of their article (what does this mean?). If published, this will include your full peer review and any attached files.

**Do you want your identity to be public for this peer review?** For information about this choice, including consent withdrawal, please see our Privacy Policy.

Reviewer #1: **Yes: **Dr Md Abul Hasan

Reviewer #2: No
